# Genome-Wide Analysis and Functional Characterization of the UDP-Glycosyltransferase Family in Grapes

Yongzan Wei [1,2,3], Huayuan Mu [1,2], Guangzhao Xu [1,2], Yi Wang [1], Yang Li [1], Shaohua Li [1] and Lijun Wang [1,*]

1 Beijing Key Laboratory of Grape Science and Enology and Key Laboratory of Plant Resources, Institute of Botany, Chinese Academy of Sciences, Beijing 100093, China; wyz4626@163.com (Y.W.); 15155189783@163.com (H.M.); guangzhao90@163.com (G.X.); wangyi19881107@163.com (Y.W.); zzllwly@126.com (Y.L.); shhli@ibcas.ac.cn (S.L.)
2 University of the Chinese Academy of Sciences, Beijing 100049, China
3 Institute of Tropical Bioscience and Biotechnology, Chinese Academy of Tropical Agricultural Sciences, Haikou 571101, China
* Correspondence: ljwang@ibcas.ac.cn; Tel.: +86-010-62836664

**Abstract:** Grape (*Vitis vinifera*) produces a variety of secondary metabolites, which can enhance nutrients and flavor in fruit and wine. Uridine diphosphate-glycosyltransferases (UGTs) are primarily responsible for the availability of secondary metabolites by glycosylation modification. Here, a total of 228 putative UGTs were identified in *V. vinifera*, and VvUGTs were clustered into 15 groups (A to O) and unevenly distributed on 18 chromosomes. Diverse VvUGT members from 12 groups were transcribed, and they responded to different external stresses. More than 72% of *VvUGT* members were expressed at one or more stages of grape fruit development, and the expression levels of 34 *VvUGT* members increased gradually with fruit ripening. The *VvUGT* members of different groups may be involved in the synthesis and accumulation of flavonoid glycosides, glycosidically bound volatiles, and stilbenes. These results will provide guidance for further research on the functions and regulating mechanisms of *UGT* genes.

**Keywords:** *Vitis vinifera*; VvUGT; fruit quality; gene expression

## 1. Introduction

Glycosyltransferases (GTs, EC 2.4.x.y) are a highly divergent family of multi-source genes, widely present in all living organisms [1]. In plants, GTs change the biological activity of small-molecule compounds by transferring glycosyl donors with receptors and then participate in and influence the growth and development of plants, formation of secondary metabolites, and response to environmental signals [2,3]. Glycosyltransferases are divided into 106 families (GT1~GT106) based on the specificity of substrates, amino acid sequence similarity, and catalytic specificity. The GT1 family contains the largest number of members, and it has the closest relationship with plants [1]. GT1 primarily uses uridine diphosphate (UDP)-glucose as a glycogen donor, which is also known as UDP-glycosyltransferase (UGT). Most UGTs are closely associated with glycosylation of important secondary metabolites, such as flavonoids, terpenoids, and sterols belonging to this family [2–4].

UGTs catalyze glycosylation to glycosyl combined with anthocyanidins, flavonols, monoterpenols, stilbenes, flavones and flavanones, polysaccharides, and other small molecules, thereby leading to the formation of fruit qualities such as color, aroma, and flavor [2,5,6]. Based on whole-genome information, the UGT gene family of some main fruit crops such as apple (*Malus × domestica*) [7], peach (*Prunus*) [8], pear (*Pyrus bretschneideri*) [9], and pomelo (*Citrus grandis*) [10] has been gradually identified. According to the location of glycosylated receptor molecules, GTs in plants can be classified into four types: O-type, N-type, S-type, and C-type. Most of the glycosyltransferases in fruits belong to the O-type, such as flavonoid 3-O-glucosyltransferase (UFGT) for the synthesis

of fruit anthocyanin [11,12], resveratrol/hydroxyl cinnamic acid O-glucosyltransferase (VlRSGT) for the synthesis of resveratrol 3-O-glucoside [13,14], and naringin-7-O-glucoside rhamnotransferase for the formation of naringin [15].

The composition and content of flavonoid glycosides, glycosidically bound volatiles (GBVs), and stilbenes directly affect the quality of the fruit and wine. UGTs have been shown to be key enzymes in the biosynthesis and accumulation of these secondary metabolites [11–14]. To date, an increasing number of UGT members have been identified and functionally characterized from grapes, such as UFGT [11,12], VlRSGT [13,14], flavonol-O-glucuronosyltransferase (VvGT5) and bifunctional 3-O-glucosyltransferase/galactosyltransferase (VvGT6) [16], phenolic acid O-glucosyltransferases (VvgGT1–3) [17], and monoterpenol β-D-glucosyltransferase including VvGT7, VvGT14a, VvGT15a, VvGT15b, and VvGT15c [6,18].

However, the genome-wide identification of the UGT family in grape has not been systematically identified, which prompted us to further clarify the potential functions of grape UGTs. Here, a total of 228 grape UGT members were isolated and identified from the *V. vinifera* genome to comprehensively explore the potential functions of grape UGTs. Phylogenetic analysis, chromosomal localization and duplication, gene structure, and conserved motifs of all UGT members were analyzed step-by-step. Subsequently, the distribution of cis-elements in promoters of *VvUGT* genes, expression profiling of *VvUGT* at different development stages of grape, and response to various abiotic stresses were investigated. These results will provide insights for future analyses of the functions and regulations of UGT in secondary metabolite formation.

## 2. Materials and Methods

### 2.1. Data Collection

The genome and assembly of *V. vinifera* were obtained from EnsemblPlants Database (http://plants.ensembl.org/index.html, accessed on 27 June 2020) [19], and the expression profile data of grape were obtained from Grape-RNA databases (http://www.grapeworld.cn/gt/, accessed on 27 June 2020). Genomic data of *Arabidopsis thaliana* were acquired from TAIR10 database (http://www.arabidopsis.org, accessed on 27 June 2020) [20], and the amino acid sequences of annotated UGTs from other plant species were downloaded from NCBI resources (https://www.ncbi.nlm.nih.gov/, accessed on 27 June 2020).

### 2.2. Identification of UGT Genes

The profile hidden Markov model (HMM) of the UGT domain (PF00201) was obtained from Pfam database 33.1 (http://pfam.xfam.org/, accessed on 21 February 2021) [21], and the putative UGT genes of *A. thaliana* were identified by HMMER V 3.3.2 web server [22]. The highly conserved protein sequences of UGT were selected from the putative *UGT* genes, which were used to build a species-specific HMM model using HMMER V 3.3.2. Then, the final *UGT* genes were isolated and identified from the grape genomic data based on the species-specific HMM profile of UGTs. Finally, Online PFAM (http://pfam.xfam.org/, accessed on 21 February 2021) and MEME Suite (http://meme.nbcr.net/meme/, accessed on 21 February 2021) were used to remove redundant sequences and isoforms and check the validation of final *UGT* genes.

In addition, Online ExPASy program (http://web.expasy.org/computepi/, accessed on 22 February 2021) was used to predict the molecular weight and isoelectric point of UGT proteins in grapes, and SignalP 4.1 Server (http://www.cbs.dtu.dk/services/SignalP/, accessed on 22 February 2021) was used to predict whether the nitrogen terminal of amino acid sequences contained signal peptides.

### 2.3. Phylogenetic Analysis Nomenclature of UGT Genes in Grape

The putative amino acid sequences of UGTs from *V. vinifera* and *A. thaliana* [23] were aligned with MUSCLE, and a phylogenetic tree was constructed using the maximum likelihood method with 1000 bootstrap replicates in MEGA 6.0 software (https://www.me

gasoftware.net/, accessed on 23 February 2021) [24]. The iTOL website (http://itol.embl.
de/, accessed on 23 February 2021) was used to visualize the completed phylogenetic tree
and describe the classification features of UGTs.

All identified *UGT* genes in grapes were classified and named according to the stan-
dards of the HUGO Gene Nomenclature Committee [25]. The name of an UGT gene was
composed of the following parts: (1) at the beginning of UGT, the gene was defined as a
member of the UGT family; (2) Arabic numbers indicated that the members of the same
number are a family, and their amino acid consistency was more than 45%; (3) letters
indicated the same subfamily, and the UGT amino acid sequence within the member of the
same subfamily was more than 60%; (4) numbers indicated individual UGT gene members.

### 2.4. Chromosomal Location and Duplication Analysis

The physical distribution and locations of the *VvUGT* genes on chromosomes were
drafted with Map Chart 2.2 software (Wageningen, The Netherlands) according to the
gene position in the genome of *V. vinifera*. Duplicate genes within the grape genome were
identified using OrthoMCL17 (Chicago, IL, USA) [26]. The microsynteny relationships
between each pair of chromosomes were analyzed using the Multiple Collinearity Scan
tool kit (MCscanx, Athens, GA, USA) [27]. Each duplicate segment with VvUGT members
was selected; the syntonic map was generated by CIRCOS (Vancouver, BC, Canada) [28],
and the duplication genes were linked by the connection lines.

### 2.5. Gene Structure and Conserved Motifs of Grape UGT Genes

The exon–intron structures of *VvUGT* genes were constructed by determining the
intron positions, intron length, 5′-untranslated regions (UTR), and 3′-UTR using TBtools
software (v1.075) (Guangzhou, China) [29]. Then, ten conserved motifs in 228 VvUGT
protein sequences were acquired and identified by the MEME Suite Wrapper module of
TBtools software [29].

### 2.6. Distribution of Cis-Elements in the Promoter of VvUGT Genes

Upstream sequences (2000 bp) from the start codon of each *VvUGT* gene were obtained
from genomic data of grape to understand the functions of the *VvUGT* genes in grapes.
Twenty-four cis-acting elements were identified on promoter sequences of 228 *VvUGT*
genes using PlantCARE software (http://bioinformatics.psb.ugent.be/webtools/plantca
re/html/, accessed on 5 April 2021) [30]. Nine elements responded to plant hormones;
eight elements were related to growth and development, and seven elements responded to
abiotic stress.

### 2.7. Expression Profiling of Grape UGT Genes

Expression profile data were derived from Grape-RNA database (http://www.gr
apeworld.cn/gt/, accessed on 27 March 2021). Transcriptome data included five grape
cultivars, namely, *V. vinifera* cv. Kyoho (KH), *V. vinifera* cv. Cortador (CT), *V. vinifera* cv.
GrosColman (GC), *V. labrusca* cv. Concord (CC), and *V. labrusca* cv. Beta (BT) was used to
investigate the expression profiles of *VvUGT* genes at different development stages of grape
fruit. Each grape cultivar had three fruit developmental stages: before mature transition
stage (F), mature transition stage (V), and ripening stage (R). Meanwhile, the RNA-Seq
data involved in UVC treatment (254 nm UVC for 10 min), cold treatment (4 °C cold for
12 h), and drought treatment (8% PEG-6000 treatment for 12 h) were used to explore the
expression profiles of *VvUGT* genes under different abiotic stresses.

The values of fragments per kilobase of exon per million reads mapped (FPKM) were
used to calculate and evaluate gene expression. The obtained FPKM values of *VvUGT*
genes were normalized with Log2, and heatmaps were plotted for further visualization
using TBtools software.

*2.8. Total RNA Isolation and qRT-PCR Analysis of Grape UGT Genes*

Total RNA was isolated from grape tissues by using a RN40 EASYspin isolation kit (Aidlab Biotechnologies, Beijing, China) in accordance with the manufacturer's instructions. The concentration and quality of each sample were determined by 1.5% agarose gel electrophoresis and BioPhotometer Plus (Eppendorf, Hamburg, Germany). RNA (2 µg) was synthesized to cDNA using the Revert Aid First-Strand cDNA Synthesis Kit (Thermo Fisher Scientific, Waltham, MA, USA) through a one-step method. qRT-PCR was performed on a LightCycler 480 II (Roche, Basel, Switzerland) using SYBR Green qPCR Master Mixes (Thermo Fisher Scientific, Waltham, MA, USA). The relative expression levels of genes were calculated using the $2^{-\Delta\Delta Ct}$ method [31]. All quantitative PCRs were performed with three biological replications. The primers used for the qRT-PCR are listed in Table S1.

## 3. Results

*3.1. Identification of UGT Family Genes in Grape*

A species-specific HMM model was constructed with 107 *Arabidopsis* putative UGT amino acid sequences, and 275 putative UGTs were extracted from the genomic data of grape based on this HMM model of UGTs. The plant secondary product glycosyltransferase (PSPG) conserved domain analysis of putative UGTs was performed by online PFAM and Meme Suite to remove redundant sequences and subtypes. Finally, a total of 228 grape UGT members were obtained containing a PSPG conserved domain at the C-terminus of the UGT proteins (Table S2).

The predicted amino acid length of the UGT protein ranged from 62 aa to 1398 aa in grapes, and the average amino acid length was approximately 453 aa. The predicted molecular weight of the UGT protein was between 6.58 and 156.67 kDa, and the isoelectric point (pI) ranged from 4.61 to 9.84. The results of subcellular localization demonstrated that 167 and 135 UGT members were predicted in the chloroplast and cell membrane, respectively. The rest of the UGT members were predicted to be located in the nucleus (15), cytoplasm (10), mitochondrion (8), and peroxisome (5) (Table S2). The signal peptide prediction results showed that except for one UGT sequence (Vit_217S0000G07071.1) with a signal peptide (Sec/SPI) value of 48.2%, all the other UGT sequences had a signal peptide (Sec/SPI) value less than 5%. The results showed that the amino acid sequence of grape UGT did not contain the signal peptide of subcellular localization, but it only encoded intracellular enzymes.

*3.2. Phylogenetic Analysis and Name of the VvUGT Family*

Phylogenetic trees were constructed between *V. vinifera* and *A. thaliana* by the neighbor-joining method to clarify the evolutionary relationship and classification of the UGT gene family. Then, 228 identified UGT members in grapes were classified and named by phylogenetic tree analysis of *A. thaliana* based on the common naming rules of the UGT gene (Figure 1). The results showed that most of the UGT members in grapes could cluster with the corresponding groups of *A. thaliana.* All grape UGT members were divided into 15 groups (A~O), and three members of Group O had no homologous genes in *A. thaliana* (Figure 1 and Table 1).

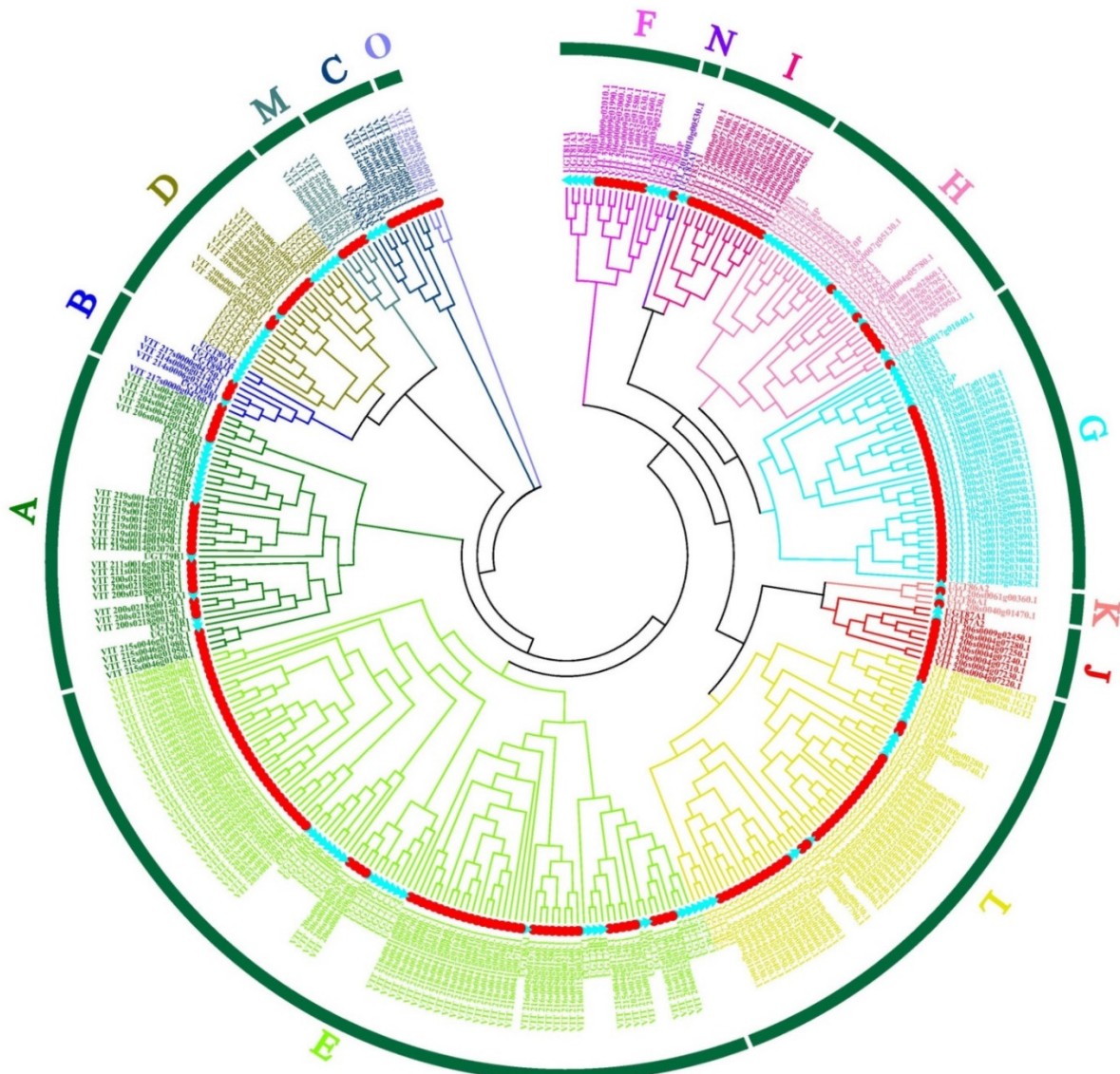

**Figure 1.** Phylogenetic analysis of UGT gene families between *V. vinifera* and *A. thaliana*. The maximum likelihood tree was constructed by MEGA 6.0 software with 1000 replications. Red and teal represent the UGT genes in *V. vinifera* and *A. thaliana*, respectively.

**Table 1.** Number of UGT members in model plants and different fruit trees.

| Species | Phylogenetic Group | | | | | | | | | | | | | | | | | References |
|---|---|---|---|---|---|---|---|---|---|---|---|---|---|---|---|---|---|---|
| | A | B | C | D | E | F | G | H | I | J | K | L | M | N | O | P | Total | |
| *Vitis vinifera* | 25 | 4 | 6 | 9 | 45 | 8 | 29 | 7 | 13 | 7 | 2 | 33 | 5 | 1 | 3 | | 228 | |
| *Arabidopsis thaliana* | 14 | 3 | 3 | 13 | 22 | 3 | 6 | 19 | 1 | 2 | 2 | 17 | 1 | 1 | | | 107 | Li et al., 2001 |
| *Oryza sativa* | 14 | 9 | 8 | 23 | 38 | | 20 | 7 | 9 | 3 | 1 | 23 | 5 | 2 | 6 | 9 | 180 | Caputi et al., 2012 |
| *Populus trichocarpa* | 12 | 2 | 6 | 14 | 49 | | 42 | 5 | 5 | 6 | 2 | 23 | 6 | 1 | 3 | 2 | 178 | Caputi et al., 2012 |
| *Malus × domestica* | 34 | | 8 | 11 | 50 | 3 | 50 | 17 | 12 | 13 | 7 | 15 | 8 | 3 | 3 | 8 | 242 | Zhou et al., 2017 |
| *Prunus persica* | 10 | 2 | 4 | 19 | 29 | 4 | 34 | 9 | 5 | 7 | 7 | 18 | 14 | 1 | 1 | 4 | 168 | Wu et al., 2017 |
| *Pyrus bretschneideri* | 5 | 4 | 2 | 8 | 31 | 6 | 33 | 10 | 10 | 2 | 9 | 1 | 10 | 3 | | 3 | 139 | Cheng et al., 2019 |
| *Citrus grandis* | 17 | 3 | 1 | 18 | 25 | 2 | 9 | 17 | 17 | 3 | 2 | 12 | 7 | 1 | 4 | 7 | 145 | Wu et al., 2020 |

To date, the UGT family of some fruit crops, such as apple (*Malus × domestica*) [7], peach (*P. persica*) [8], pear (*P. bretschneideri*) [9], and pomelo (*C. grandis*) [10], have been reported. The UGT genes in grapes contained 228 members and greatly outnumbered model plants such as *A. thaliana* (107), rice (180), and poplar (178) (Table 1). Compared with other fruit trees, the number of UGT genes in grapes was more than that in peach (168), pear (139), and pomelo (145) but less than that in apple (242; Table 1). At present, a total

17 groups of UGTs have been found and identified in angiosperms [9]. A total of 15 groups of UGTs in grapes were detected, and Groups P and Q were missing. Groups E, L, G, and A had the highest number of VvUGT members, which were 45, 33, 29, and 25, respectively. In grapes, the number of UGT members in Group L, I, and F was higher than those of all other species (Figure 1 and Table 1), showing that VvUGT in these three groups showed significant gene expansion. The expansion of UGT members may be closely related to the variety and rich content of secondary metabolites in grape fruits.

*3.3. Chromosomal Distribution and Gene Duplication of VvUGT Genes*

Chromosomal localization analysis of 228 *VvUGTs* was performed to understand the distribution of *UGT* genes on the grape genome (Figure 2). The results showed that except for Chr10, 197 *VvUGT* genes were distributed unevenly on 18 grape chromosomes. Seventy-six *VvUGT* genes were distributed on chromosomes Chr05, Chr18, and Chr13, which accounted for about one-third of the total number of *VvUGT*. In addition, evident clusters were distributed on 12 chromosomes, such as Chr05, Chr18, Chr13, and Chr12 (Figure 2).

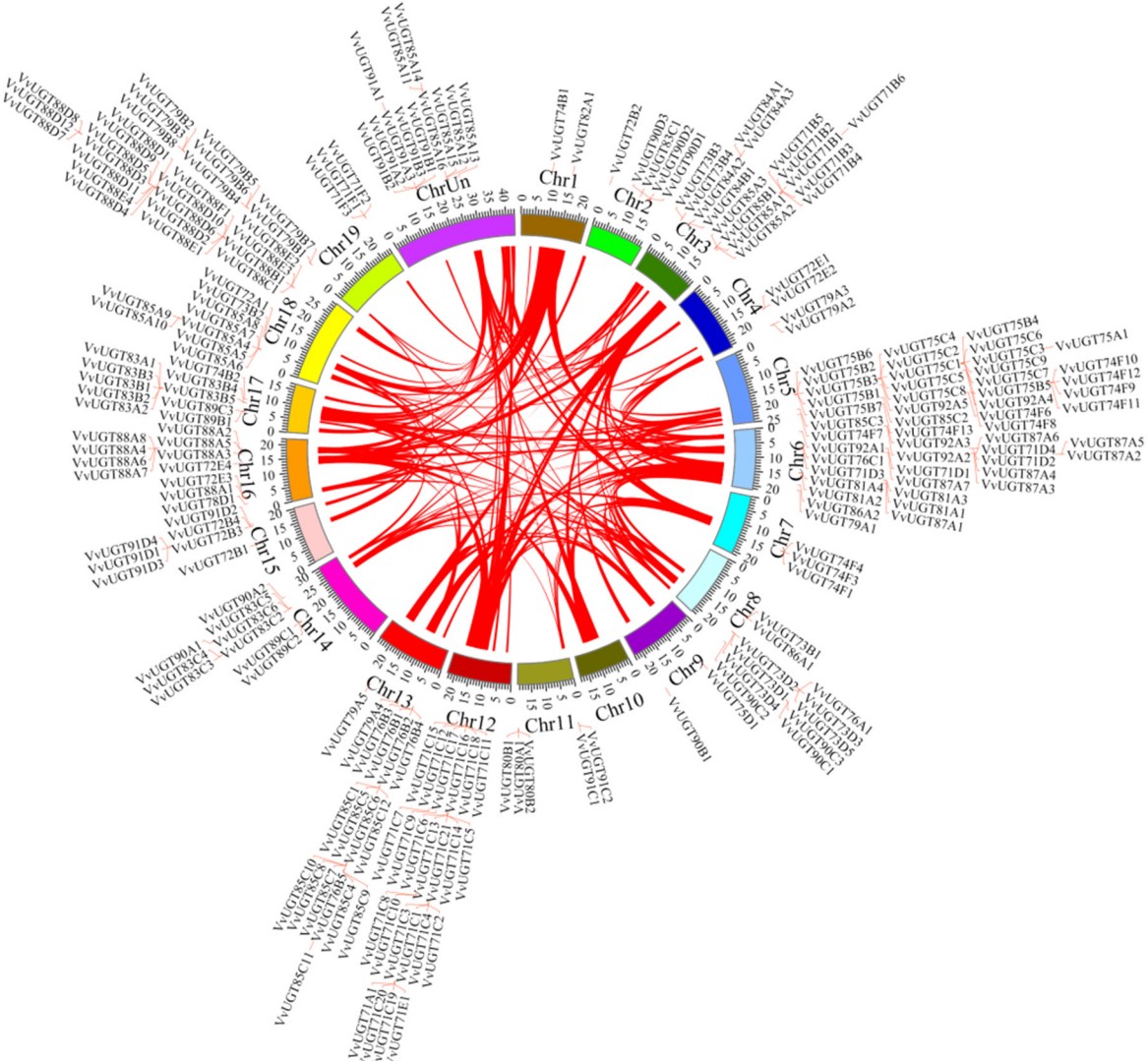

**Figure 2.** Chromosomal distribution and duplication analysis of UGT genes in grapes. UGT genes were mapped into different chromosomes in grapes. The chromosome number is indicated on the outside. Red lines connecting two chromosomal regions indicate duplicated regions among grape chromosomes.

Gene duplication events were important for gene family expansion, which commonly occurred in plant evolution [32]. Gene duplications and tandem repeat events of *VvUGT* were systematically analyzed to explain the role of gene duplication in the formation of the *UGT* gene family in grapes (Figure 2). The results showed that the 163 *VvUGT* genes were involved in 1533 fragment duplication events; thus, the expansion of the UGT family may be improved by the duplication events of chromosomal fragments. In addition, analysis of tandem repeats revealed that 108 *VvUGT* genes were involved in tandem repeat events, which generated the paralogous gene cluster in the genome (Figure 2). Glycation is an important reaction to modify the secondary metabolites such as terpenoids, sterols, flavonoids, phenols, and alkaloids [3–6]. The rapid expansion of the UGT family in grapes leads to similar functional enhancement and functional diversity, which may cause the rich variety of secondary metabolites in grapes.

### 3.4. Analysis of the Conserved Motif of VvUGT Genes

The number and distribution of functional domains on protein sequences can reveal the structural similarities and differences among gene family members, which can be used as an important basis for phylogenetic classification of gene families. Ten domains of 228 VvUGT amino acid sequences were analyzed using the Meme Suite Wrapper module in TBtools software to elucidate the motif characteristics of the UGT gene family in grapes (Figure 3). Subsequently, the protein domain database (SMART) was used to annotate the 10 selected domains, which showed that Motif 1, Motif 2, and Motif 3 were annotated as the UDP-glucuronosyltransferase (UDPGT) domain (PF00201). Motif 1 contained complete PSPG-box. A total of 210 VvUGT members contained all three motifs of UDPGT, whereas 16 VvUGTs contained two UDPGT motifs, and two VvUGTs contained one UDPGT motif (Figure 3). This finding indicated that the identification of the VvUGT gene family in grapes was reliable.

The number of motifs ranged from 1 to 10 in VvUGT sequences, and more than 50% (126) of the VvUGT members contained all 10 motifs (Figure 3). In most VvUGT sequences, Motif 4 was located at the C-terminal of the UGT sequence, and Motif 9 was positioned in the N-terminal of the UGT sequence. All members of Group A and most members of Group C did not include Motif 9 (Figure 3). The differences in the number and distribution of these motifs may be related to the functional differentiation of UGT members.

### 3.5. Analysis and Gene Structure of VvUGT Genes

The characteristics of the gene structure are an important basis for phylogenetic and taxonomic analysis [33]. The UDPGT conserved domain, intron, exon, and UTR of the *UGT* gene were analyzed in detail by using TBtools software to explore the evolutionary relationship of VvUGT [29]. All 228 *VvUGT* members had UDPGT conserved domains; 42.1% (96) of the *VvUGT* sequences contained one or more introns, and 57.9% of the VvUGT members lost introns (Figure 4). In class clusters, all *VvUGT* members of Groups F, N, and K had one or more introns, but all *VvUGT* members of Groups O and B lost introns (Figure 4).

The results of 5′-UTR and 3′-UTR analysis showed that 114 VvUGT member sequences contained 5′-UTR and 3′-UTR; 19 member sequences contained only 5′-UTR, and 53 member sequences included only 3′-UTR (Figure 4). All members of Groups N, J, and K and 36 members of Group E contained 5′-UTR and 3′-UTR. Among the 42 *VvUGT* members that lost UTR, Groups L and E had the largest number of members, reaching 9 and 8, respectively (Figure 4).

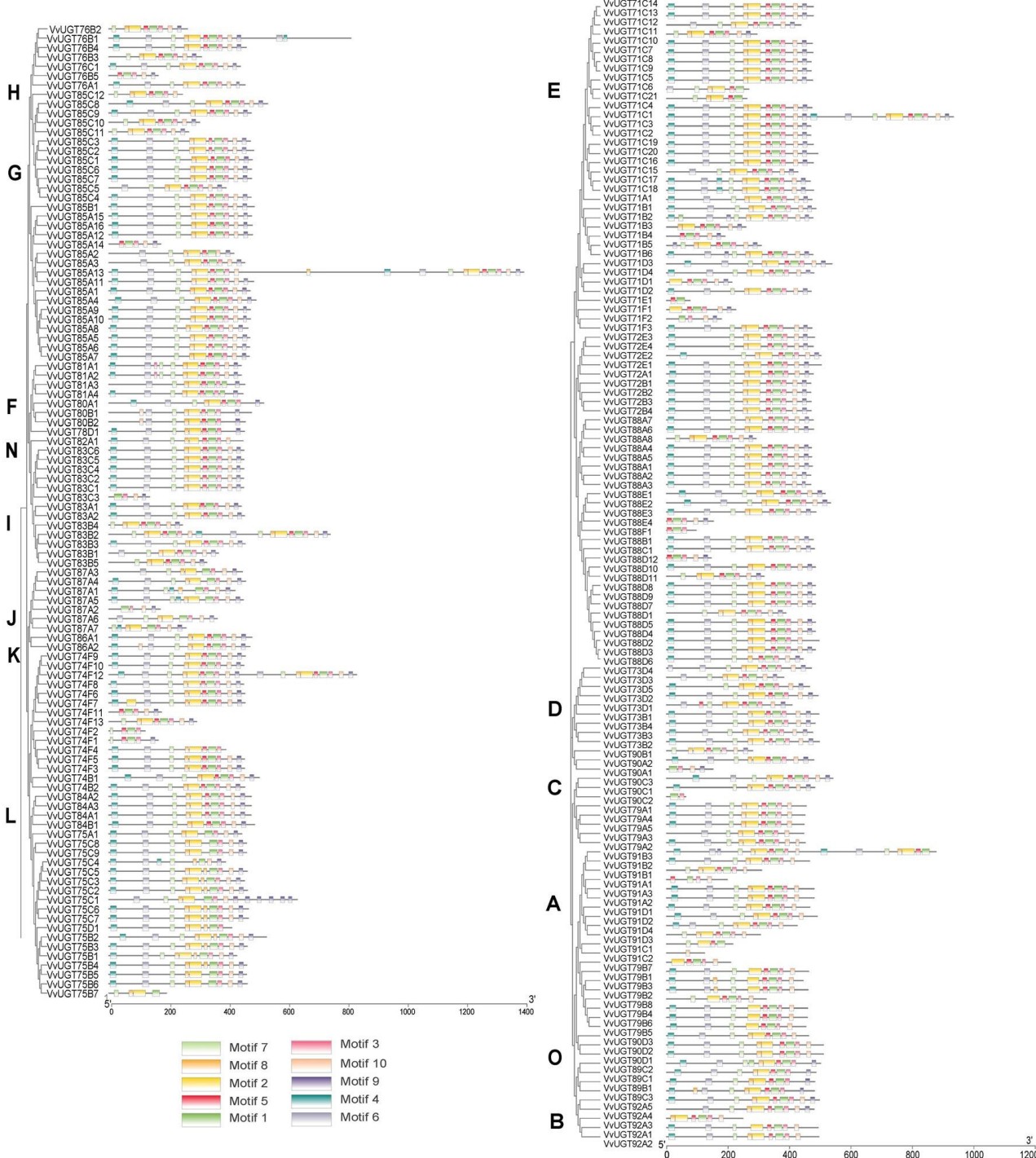

**Figure 3.** Analysis of conserved motifs of *UGT* members in grapes.

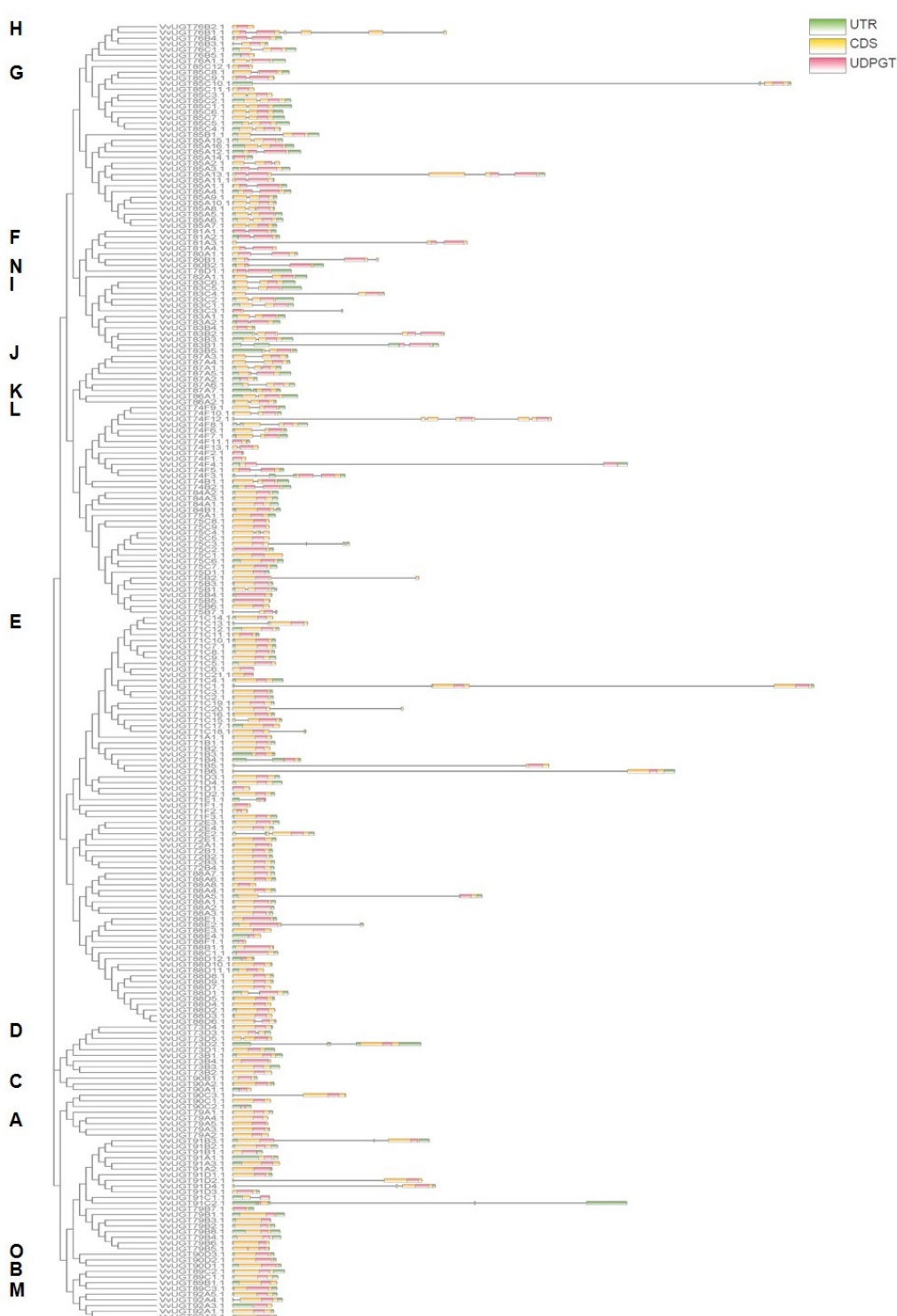

**Figure 4.** Analysis of gene structures among *UGT* genes in grapes.

### 3.6. Analysis of Cis-Elements in the Promoters of VvUGT Genes

Twenty-four types of cis-acting elements in *UGT* promoters were computed and compared using PlantCare online database to understand the distribution characteristics of cis-acting elements in the promoter sequence of *VvUGT* (Figure 5). These cis-acting elements can be divided into three categories: plant hormone, plant growth and development, and abiotic stress. Plant hormone-responsive elements primarily included TGA-element and AuXRR-core (auxin); TATC-box, P-box, and GARE-motif (gibberellin); TCA-element (salicylic acid); ABRE (abscisic acid); CGTCA-motif (jasmonic acid); and ERE (ethylene).

Except for *VvUGT91B1/D4* and *VvUGT88D9*, all other *VvUGT* promoter sequences had one or more plant hormone-responsive elements ranging from 1 to 20 (Figure 5).

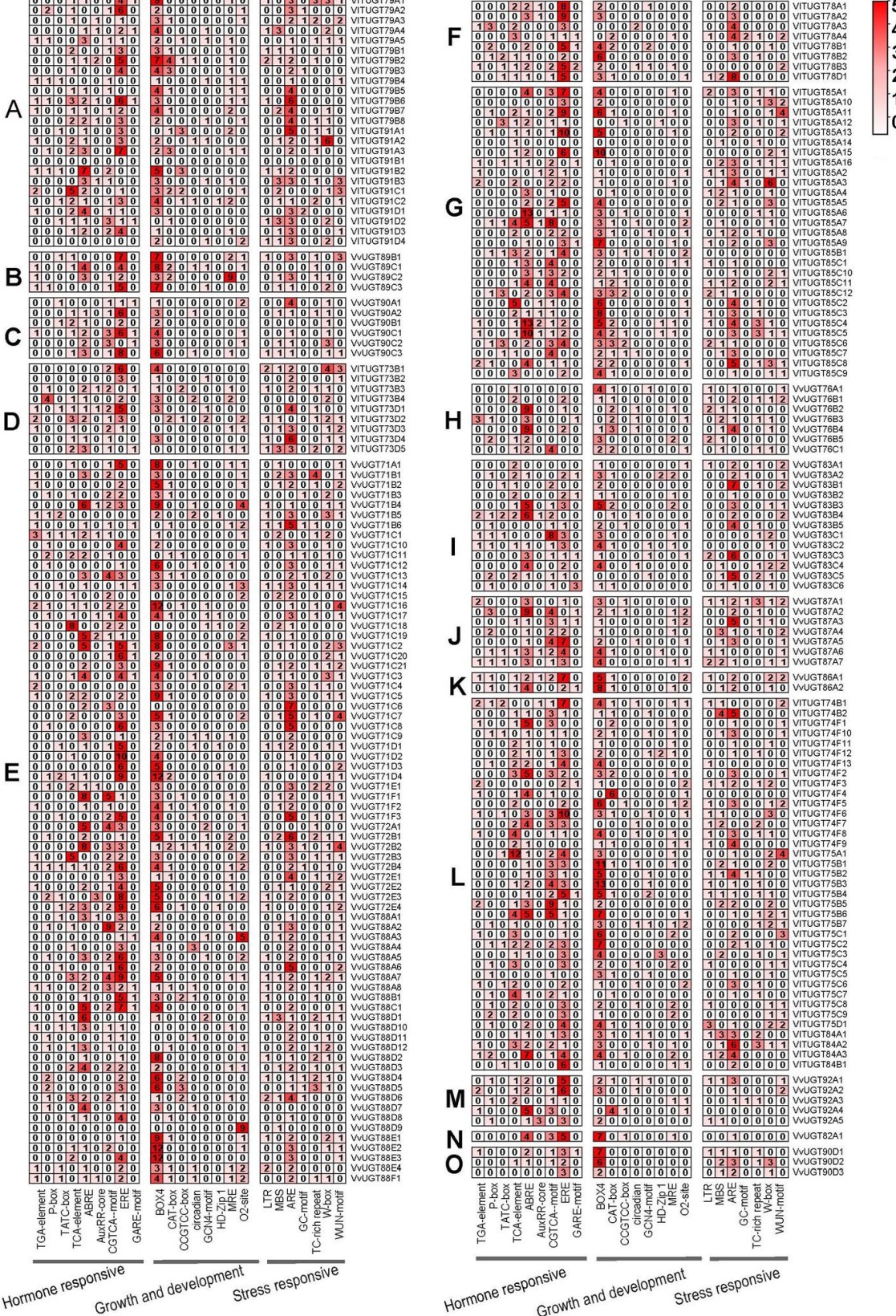

**Figure 5.** Predicted cis-acting elements in the promoter regions of *VvUGT* genes. The number inside the boxes indicates the number of cis-acting elements.

The promoter sequences of *VvUGT85A7*/C4 and *VvUGT75A1* contained 20 hormone-responsive elements. ERE was the most widely distributed element, which existed in 78.5% of UGT members, among which the promoters of *VvUGT85A13* and *VvUGT74F6* contained 10 ERE elements (Figure 5). This result indicated that UGT may be induced by auxin, gibberellin, salicylic acid, abscisic acid, ethylene, and jasmonic acid, and it was closely related to ethylene signaling.

The cis-acting elements related to plant growth and development primarily involved BOX, CAT-box, CCGTCC-box, circadian, GCN4-motif, HD-Zip1, MRE, and O2-site. Except for *VvUGT91B1*, *VvUGT71C6*, and *VvUGT85A14*, the promoter sequences of all other *VvUGT* members contained one or more cis-acting elements related to plant growth and development. BOX4 was the most widely distributed element and was abundant in *VvUGT* promoters, with 86% of *VvUGT* members containing BOX4 elements and 57.5% of *VvUGT* members containing 3 or more BOX4 elements (Figure 5).

The cis-acting elements in response to abiotic stresses primarily included LTR (low temperature), MBS (drought), ARE (anaerobic sensing), GC-motif (anaerobic), TC-rich repeat (defense and stress), W-box (stress resistance), and WUN-motif (trauma). In grapes, 97.8% of *VvUGT* promoter sequences contained one or more cis-acting elements that were responsive to external stresses. ARE is the most widely distributed and abundant in *VvUGT* promoter sequences, with 81.1% of *VvUGT* members having ARE elements, and 34.2% of *VvUGT* members having 3 or more ARE elements (Figure 5). In addition to *VvUGT91B1*, *VvUGT88D7*, and *VvUGT92A4*, all other *VvUGT* promoter sequences contained one or more cis-acting elements in response to abiotic stresses, among which the number of *VvUGT72B1*, *VvUGT73B1*, and *VvUGT85A3* was up to 12 (Figure 5).

*3.7. Expression Profiles of VvUGT Genes in Fruits at Different Developmental Stages*

The expressions of *UGT* in five grape cultivars, including "Kyoho" (KH), "Cortador" (CT), "GrosColman" (GC), "Concord" (CC), and "Beta" (BT), were obtained from Grape RNA database to understand the expression patterns of UGT members during grape development. More than 72% of *VvUGT* members were expressed at one or more development stages of grape fruit. The expression levels of the *VvUGT* members of Group C were lower, and some *VvUGT* members of the other 14 groups were highly expressed at specific stages of fruit development. The expression levels of all *VvUGT* members of Groups N and O were high in the three development stages of the five cultivars. Some members of other groups were also highly expressed in the three development stages of grape fruit, such as *VvUGT89C3* of Group B; *VvUGT73B3* of Group D; *VvUGT88A1/A2/A5* of Group E; *VvUGT83C4* of Group I; *VvUGT84A2/A3*, *VvUGT74B2/F3/F8*, and *VvUGT75A1/D1* of Group L; and *VvUGT95A1/A4/A5* of Group M (Figure 6). Therefore, multiple UGT members in 14 groups were involved in the development of grape fruit, and different UGT members played different roles in specific developmental stages.

During the development of grape fruit, the expression levels of 34 *VvUGT* members increased with fruit ripening, such as *VvUGT79A1* and *VvUGT91B1* (Group A); *VvUGT89C3* (Group B); *VvUGT73B3/D2* (Group D); *VvUGT88A6/C1/D2/E1/E2*, *VvUGT71B4/C7/C8/C9/C10/C19*, and *VvUGT72E1/E4* (Group E); *VvUGT78D1* (Groups F); *VvUGT85A6/A7/A8/A9/A10/A16* (Group G); *VVUGT87A6* (Group J); *VvUGT75D1/A1*, *VvUGT74B2/F3/F8*, and *VvUGT84A2/A3* (Group L); and *VVUGT82A1* (Group N; Figure 6). Therefore, the UGT members of Groups E and L might be closely related to fruit ripening or accumulation of secondary metabolites. By contrast, the expression levels of 40 *VvUGT* members of 8 groups were higher before fruit maturation, but such expression levels gradually declined with fruit ripening (Figure 6). These UGT members may play an important role in the early developmental stages of grape fruit.

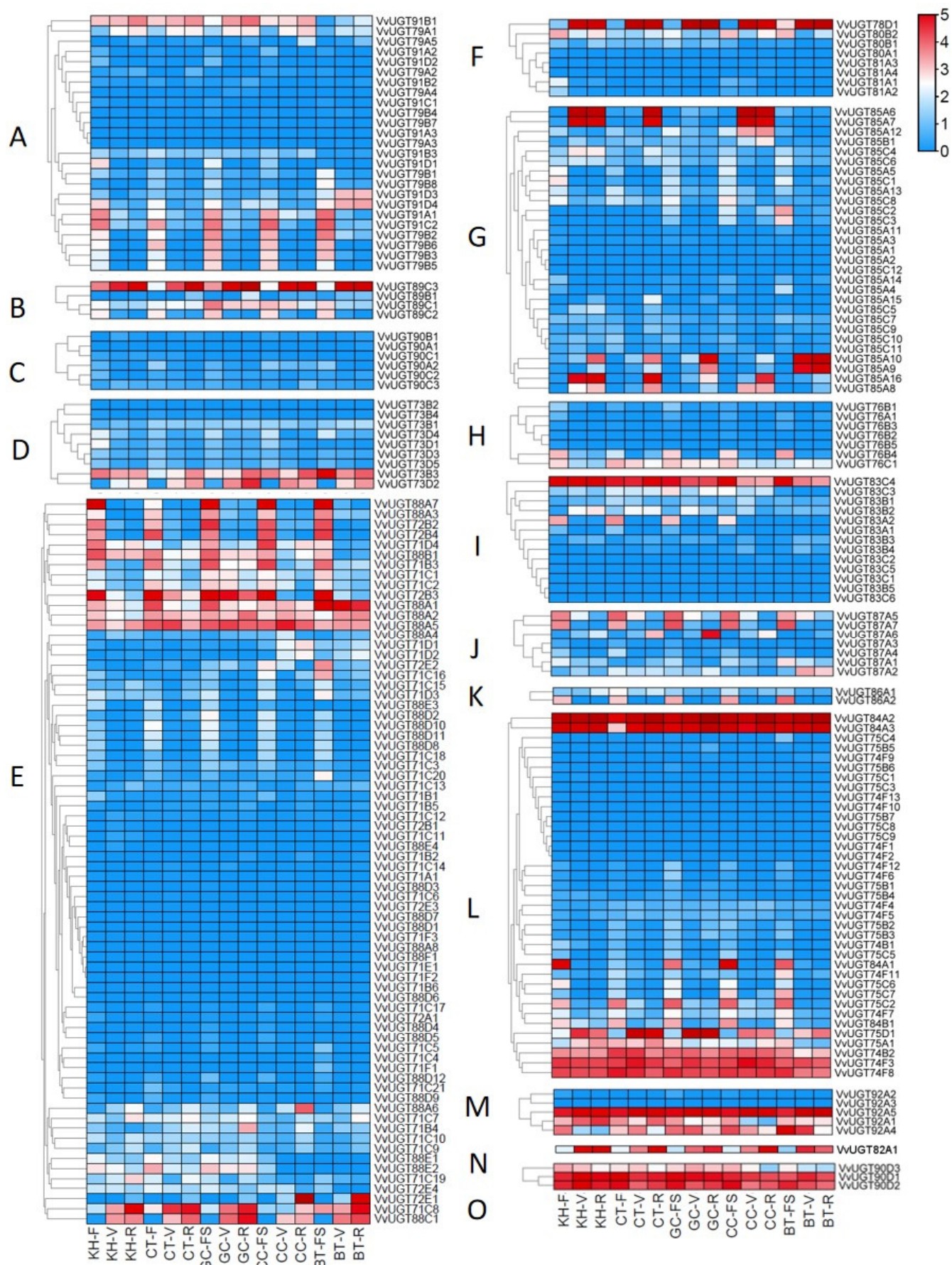

**Figure 6.** Expression profiles of *UGT* members at different development stages of grape fruit.

### 3.8. Expression Profiles of VvUGT Genes when Responding to Abiotic Stress Treatments

The expression data of *UGT* members treated with UVC, low temperature, and drought stress were selected from Grape RNA database to clarify the expression characteristics of UGT members in grape fruits when responding to abiotic stresses (Figure 7). Based on the overall expression pattern of *UGT* members, except for Groups C, H, and N, the *VvUGT* members of the other 12 groups were involved in the response to the three abiotic stresses in varying degrees (Figure 7). Low temperature affected the expression changes of *VvUGT* of 10 groups, and drought induced differences in the *VvUGT* transcriptional levels of 13 groups; moreover, UVC treatment affected the expression changes in *VvUGT* members of 7 groups (Figure 7).

After low-temperature treatment in grape fruits, the transcriptional level of 15 *VvUGT* members significantly increased (Figure 7), such as *VvUGT91D3* (Group A); *VvUGT89B1/C3* (Group B); *VvUGT71C8/C12/C14/C16/C19* and *VvUGT72B2/B3/B4* (Group E); *VvUGT80B2* (Group F), *VvUGT75C7* and *VvUGT74B1* (Group L); and *VvUGT92A5* (Group M). However, the number of *VvUGT*, of which the expression level decreased gradually, reached 30, which increased by two times the number of the expression. The number of downregulated *VvUGT* members and their groups after low-temperature treatment was more than that of upregulated *VvUGT* members (Figure 7), indicating that low temperature significantly inhibited the transcription of *VvUGT* members. *VvUGT* was directly involved in the formation of fruit quality. Therefore, low temperature may delay the formation of fruit quality such as flavonoids and aroma by inhibiting the expression of *VvUGT* members.

After drought stress treatment, the transcription levels of 26 *VvUGT* members of 7 groups in grape fruits significantly increased, whereas the expression levels of 31 *VvUGT* members of 12 groups evidently decreased (Figure 7). Thus, the number of downregulated *VvUGT* members and groups was more than the upregulated ones. Therefore, drought stress promoted the biosynthesis of some secondary metabolites and affected the formation of fruit quality. After UVC treatment, the expression levels of 16 *VvUGT* members of 7 groups were significantly increased (Figure 7). Compared with low temperature and drought stress, UVC treatment had a weaker effect on the expression of *VvUGT* family members probably because the treatment time was too short to affect the glycosylation of secondary metabolites.

### 3.9. Verification of the Gene Expression of VvUGT during Fruit Development

qRT-PCR and transcriptome data were used to compare and analyze the expression patterns of 15 *VvUGT* genes in fruit developmental stages of 'Dakeman' grapes and to clarify the roles of *VvUGT* genes during fruit development (Figure 8). The results indicated that the expression levels of 15 *VvUGTs* increased gradually with fruit development, but some differences were observed when they reached the peak value. The expression levels of 11 *VvUGT* peaked at 120 DAA (Days after anthesis, DK6) (Figure 8). During fruit ripening, the fruit flavor thickened, and the content of functional components gradually increased, suggesting that *VvUGT* may be involved in the formation of fruit quality and accumulation of functional components. In addition, the results of qRT-PCR were basically consistent with the trend of transcriptome expression, indicating that the experimental results were reliable and effective.

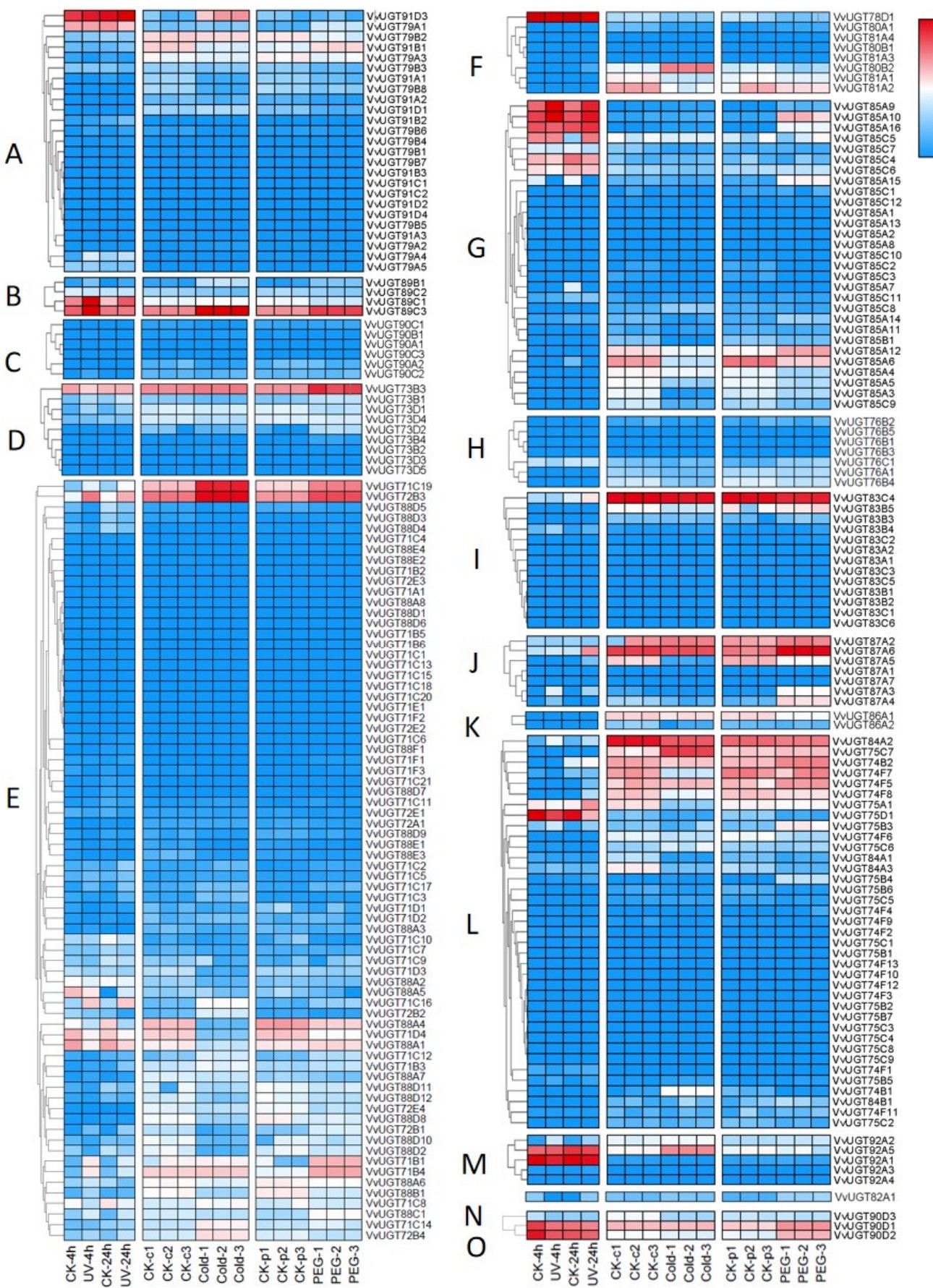

**Figure 7.** Expression profiles of *UGT* members in grapes in response to different abiotic stresses.

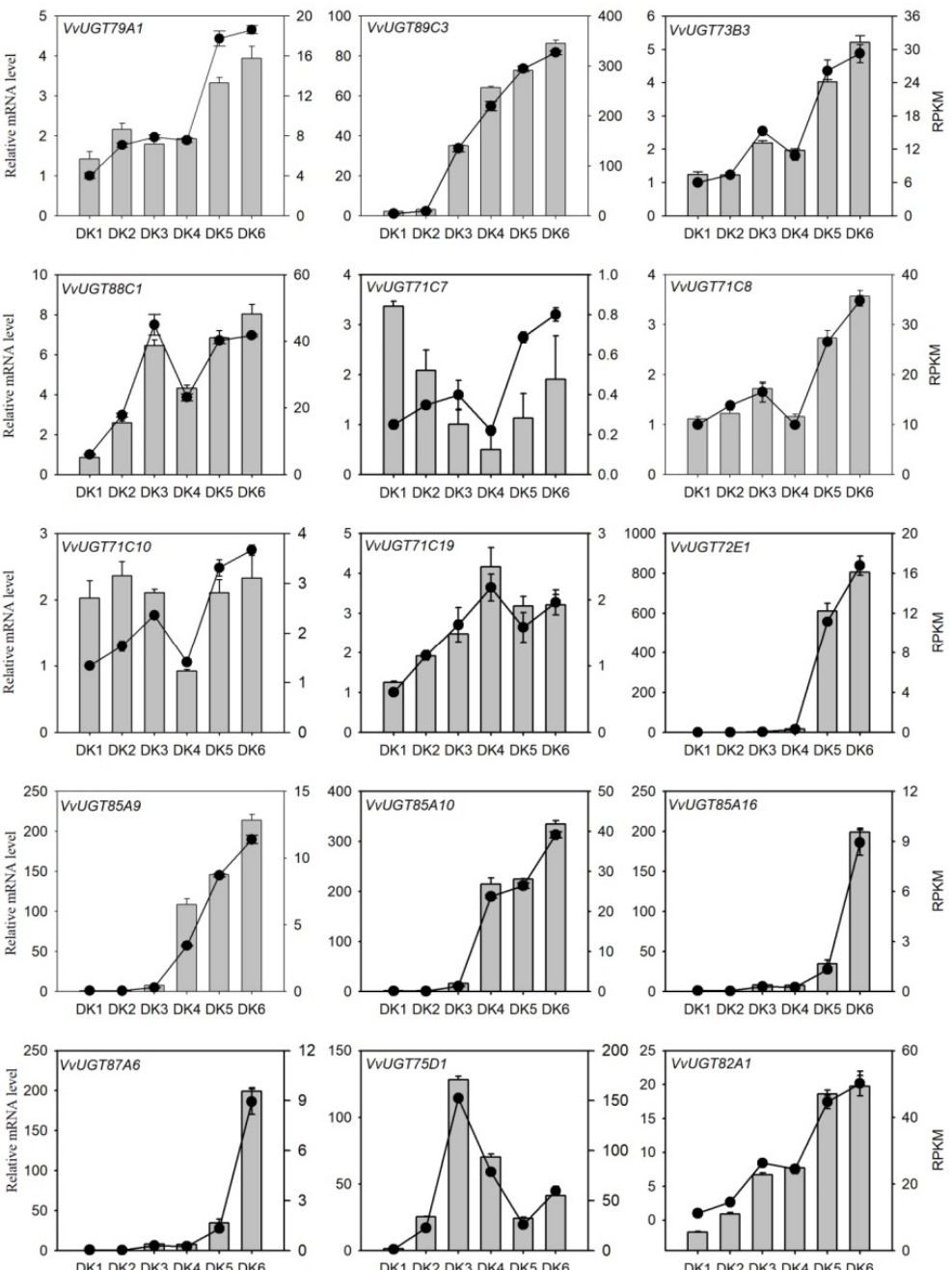

**Figure 8.** Analysis of gene expression characteristics of *UGT* members at different development stages of grapes. DK1~6 represented the developmental stages of grape fruit, 45 DAA (Days after anthesis), 60 DAA, 75 DAA, 90 DAA, 105 DAA and 120 DAA, respectively.

### 3.10. Functional Prediction of VvUGT Genes Based on Known UGT

The functions of 34 VvUGT, which upregulated during fruit ripening, were predicted compared with 26 UGTs that had their function confirmed in fruit crops by multiple sequence alignment and phylogeny evolution analysis. UGTs of known functions were primarily related to flavonoids, GBVs, and stilbenes, which affected fruit quality and flavor. The phylogenetic analysis indicated that except for VvUGT82A1 and VvUGT89C3, the other 32 VvUGT members can be clustered with UGTs of known functions (Figure 9).

Flavonoids are important secondary metabolites in grape fruits, and their content directly affects the appearance and internal quality of fruits. Phylogenetic analysis showed that the UGT members of Group A (UGT79 and UGT91), Group D (UGT73), Group F (UGT78), and Group J (UGT87) might be involved in the biosynthesis and accumulation of

flavonoids. VvUGT78D1 clustered with VvUFGT, VaUFGT, LcUFGT, and AcUFGT and was directly involved in the biosynthesis of anthocyanin. VvUGT87A6 had a close relationship with FcCGT and CuCGT in citrus, which may be related to the accumulation of 2-hydroxyflavanone, dihydrochalone, mono-C-glucoside, and other substances. VvUGT79A1 and VvUGT91B1 clustered with 1,2-RHAT and 1,6-RHAT, respectively, suggesting that they may be related to the glycosylation of flavanone 7-O-glucoside. VvUGT73B3 and VvUGT73D2 may be involved in the biosynthesis of kaempferol, quercetin, and isorhamnetin (Figure 9).

The content of volatile compounds in grape fruits directly affected fruit quality and wine flavor, and UGT played an important role in controlling the composition and content of aroma components and balancing bioactive metabolites in organisms. As shown in Figure 9, 19 UGTs belonging to Group E (UGT72, UGT88, and UGT71) and Group G (UGT85) were clustered with UGTs, which participated in volatile biosynthesis, indicating that these two groups of UGT may be involved in the GBV accumulation of grape fruits.

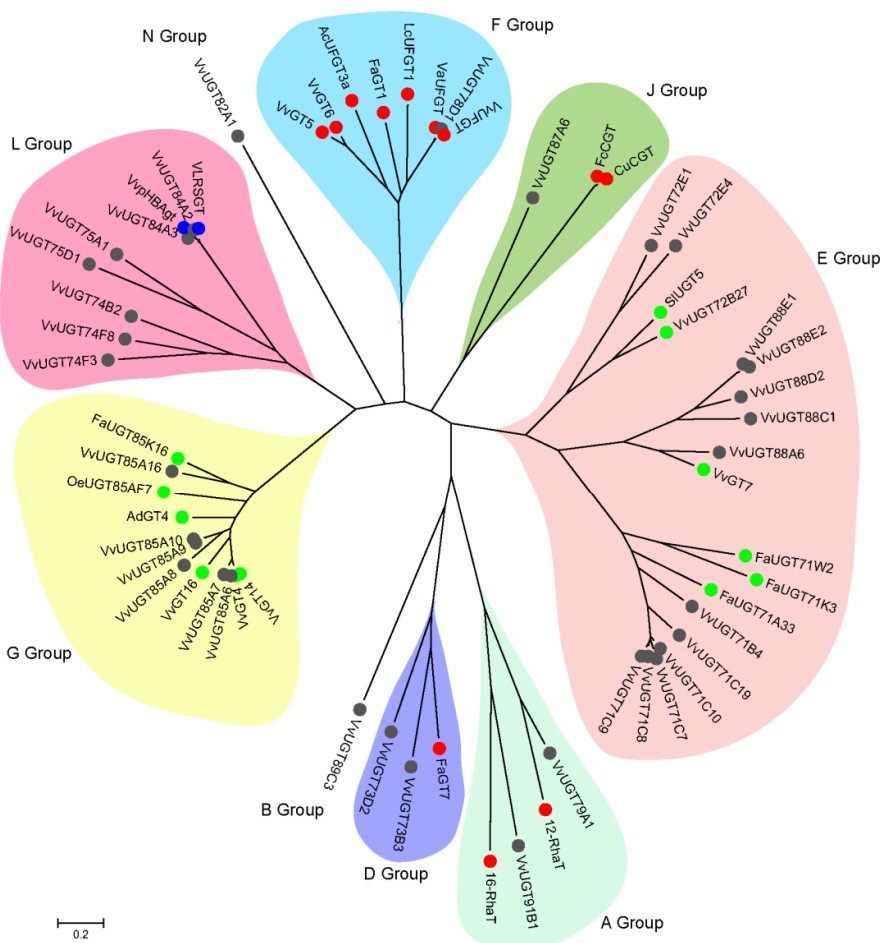

**Figure 9.** Phylogenetic analysis and functional classification of grape UGTs with other fruit trees. Grey dots represent grape UGTs. Red dots, green dots, and blue dots represent UGTs with identified functions related to the biosynthesis of flavonoids, GBVs, and stilbenes, respectively.

UGT is a key enzyme specifically catalyzing the biosynthesis of resveratrol glycosides, and VLRSgt is involved in the accumulation of stilbene in grape fruits [12,13]. Phylogenetic analysis indicated that UGT84, UGT75, and UGT74 of the L Group may be involved in the glycosylation of stilbenes (Figure 9). UGT84A2/A3 clustered with VLRSgt, and it was most related to the accumulation of stilbenes. Furthermore, UGT75A1/D1 and UGT74B2/F8/F3 may be involved in the biosynthesis of stilbenes.

## 4. Discussion

The UGT family is a superfamily involved in the glycosylation and modification of plant secondary metabolites, and it plays an important role in the formation of fruit quality such as anthocyanins, GBVs, and stilbenes [3–5,8]. With the completion of the whole genome mapping, the UGT whole gene family in model plants such as *A. thaliana* [34] and rice [35] has been identified, whereas the UGT family in major fruit crops such as apple [7], peach [8], pear [9], and pomelo [10] has also been studied sequentially. Grape is rich in flavonoids, resveratrol, and other functional ingredients, and it is cultivated worldwide. Previous studies have analyzed the role of UGT members in the metabolism of anthocyanins [11,12], stilbenes [13,14], terpenes, and terpene alcohols [6] from the perspective of metabolism, but the genome-wide systematic study on the UGT gene family of grapes has not been performed.

A total of 228 UGT members were identified in grapes, which were divided into 15 groups (A~O; Figure 1 and Table 1). The number of UGT family members of grapes was higher than that of model plants such as *A. thaliana* (107) [23], rice (180) [34], poplar (178) [34], peach (168) [8], pear (139) [9], and pomelo [10], and less than only that of apple (242) [7] (Table 1). Highly conserved Groups A~N in plants [34] were found in five major fruit crops. Groups O and P were found in the UGT family of apple, peach, and pomelo; only Group O was discovered in grapes, and only Group P was found in pears. Group Q was not found in five major fruit crops, but it was found in maize [36] and wheat [37], indicating that fruit crops lost the UGT members of Group Q during plant genome evolution. The number of UGT members in Groups E, L, G, and A was more than 20, and UGT members of Groups L, A, and I showed significant gene expansion compared with other species. Therefore, grape fruits had abundant secondary metabolites, which may be related to the expansion of these UGT members.

Gene duplication events are ubiquitous in plant evolution, with more than 70% of angiosperms experiencing replication events, which can cause the expansion of gene families [34]. With regard to the distribution of *UGT* genes in grape chromosomes, except for Chr10, the UGT genes were unevenly distributed in 18 other chromosomes. A total of 1533 gene duplicative events involved in 163 *UGT* genes occurred in the UGT family, and 108 *UGT* members experienced tandem repeat events (Figure 2). Therefore, the expansion of the *UGT* family may be completed with the duplication events of chromosomal fragments, and tandem repeat events may cause the expansion of the *UGT* gene. Segmental duplication events contribute to the ancient duplication of the *UGT* family, and tandem duplication leads to new expansion of UGT families. Glycation is an important step in modifying secondary metabolites [2,5,6]. The rapid expansion of the UGT family in grapes leads to functional diversity and enhances similar functions, which may result in the rich variety of secondary metabolites such as flavonoids, terpenes, and stilbenes in grape fruits.

Gene expression in higher plants is primarily regulated by the coordinate regulation of cis-acting elements and transcription factors, and the cis-acting elements of the promoter initiate and determine the direction and efficiency of specific transcription [38]. Therefore, understanding the characteristics of cis-acting elements of the promoter is the premise for analyzing the characteristics of gene transcriptional regulation and expression patterns. In grapes, the promoters of *UGT* members contained various numbers of cis-acting elements. ERE responding to the ethylene signal was the most widely distributed element and was present in 78.5% of *VvUGT* members, among which the number of ERE elements in *VvUGT85A13* and *VvUGT74F6* promoters reached 10 (Figure 5), indicating that the expressions of the *VvUGT* gene were induced by the ethylene signal. In addition, more than 98% of the UGT member promoters contained cis-acting elements related to growth and development, and BOX4 was the most widely distributed element, with 57.5% of *UGT* members containing 3 or more BOX4 elements. Furthermore, 97.8% of the *VvUGT* member promoters contained one or more cis-acting elements in response to abiotic stresses (Figure 5). This result indicated that the transcriptional regulations of *VvUGT* genes were jointly completed by the external environment, plant hormones, and growth and development.

Analysis of expression profile data during fruit development showed that more than 72% of *VvUGT* members were expressed at one or more stages of fruit development, and multiple members of 14 groups showed high expression at specific stages of fruit development (Figure 6). The expressions of 34 *VvUGT* members of 9 groups gradually increased with fruit ripening, whereas the transcriptional levels of 40 *VvUGT* members of 8 groups were higher before fruit maturation, which gradually declined with fruit ripening (Figure 6), suggesting that UGT members may play a different role in different stages of fruit development. With regard to the overall expression pattern of *VvUGT*, 10, 13, and 7 groups of UGT members were involved in the response to low temperature, drought, and UVC (Figure 7). Therefore, VvUGT can respond to abiotic stresses such as low temperature, drought, and UVC, and different UGT groups and members were induced by different stress signals. Therefore, these three abiotic stresses promoted the synthesis of some secondary metabolites, but they also had a certain effect on the formation of fruit quality.

UGT is primarily involved in the accumulation of flavonoids, terpenes, and stilbenes through glycosylation modification, thereby affecting fruit color, aroma, and flavor [2,5,39]. The composition and content of flavonoids directly affect the appearance and internal quality of fruit, and glycosylation serves as a crucial regulator of the plant phenylalanine pathway [39]. UGT cluster analysis with known functions found that VvUGT78D1 was associated with grape VvUFGT [11,12], lychee LcUFGT [40], and kiwifruit AcUFGT [41] (Figure 9). Therefore, VvUGT78D1 may be involved in the biosynthesis of anthocyanin in grapes.

VvUGT87A6 is closely related to the evolution of kumquat FcCGT and Wenzhou mandarin CuCGT [42], and it may be related to the accumulation of 2-hydroxyflavanone, dihydrochalone, and mono-C-glucoside. VvUGT79A1 and VvUGT91B1 correlated with pomelo 1,2-rhat [43] and sweet orange 1,6-rhat [44,45], suggesting that they may be related to the glycosylation of flavanone 7-O-glucoside. VvUGT73B3 and VvUGT73D2 may be involved in the biosynthesis of kaempferol, quercetin, and isorhamnetin [46]. Therefore, the members of UGT79 and UGT91 of Group A, UGT73 of Group D, UGT78 of Group F, and UGT87 of Group J may be involved in the accumulation of different types of flavonoid.

The composition and content of volatile compounds in grape fruits directly affected fruit quality and wine flavor [18]. Six VvUGT members of Group G (UGT85) can be associated with grape VvGT14, VvGT4, and VvGT16 [18]; strawberry FaUGT85K16 [47]; and kiwifruit AdGT4 [48], which control volatile synthesis (Figure 9). These results indicated that UGT85 was closely related to the biosynthesis of citronellyl glucoside, geranyl glucoside, and neryl glucoside, and it may also be involved in the accumulation of monoterpenyl glucosides [49]. Similar to the previous conclusion, UGT85 is most closely related to the catalytic biosynthesis of volatile substances in the GT1 supergene family [49]. In addition, UGT72 and UGT88 in Group E clustered with tomato SlUGT5 [50] and grape VvGT7 [18], and UGT71 was closely related to FaUGT71W2, FaUGT71K3, and FaUGT71A33 in strawberry [47] (Figure 9). This result indicated that UGT72 may be involved in the synthesis of arbutin and β-isosalicin, and UGT88 is closely related to the formation of geranitin glucoside, and UGT71 may participate in the accumulation of furaneoyl glucoside [18,47,50]. Therefore, the UGT members of Group G (UGT85) and Group E (UGT72, UGT88, and UGT71) may play important roles in the biosynthesis of GBV in grape fruits.

VLRSgt is a specific catalytic enzyme for resveratrol glycosides that is involved in the accumulation of stilbenes in grape fruits [13,14]. According to the phylogenetic classification of UGT, UGT84A2/A3 has the closest relationship with the evolution of VLRSgt and stilbene biosynthesis. UGT75A1/D1 and UGT74B2/F8/F3 may also be involved in the glycosidylation of stilbenes. Therefore, UGT84, UGT75, and UGT74 of the L group may be closely related to the accumulation of stilbene in grape fruits.

The functional model of UGT involved in flavonoids, GBVs, and stilbenes in fruits was constructed on the basis of the UGT genes of grapes and other fruit trees [13,14,41–50] (Figure 10). First, when plant fruits are developed to a certain stage, the expression of

different UGT members can be activated under the co-regulation of external environmental signals (cold, drought, UVC, etc.) and endogenous plant hormones (ethylene, ABA, etc.), thereby affecting the formation of fruit quality such as color, aroma, and functional components. The members of UGT79/91/73/78/87 were primarily involved in the accumulation of flavonoids such as anthocyanin and flavonol [41–46], and the members of UGT85/72/88/71 were primarily involved in the biosynthesis of terpenes, citronellyl glucoside, and monoterpenyl glucosides, which controlled volatiles [18,47–50]. Furthermore, UGT84/74/75 might be involved in the formation of polydatins [13,14].

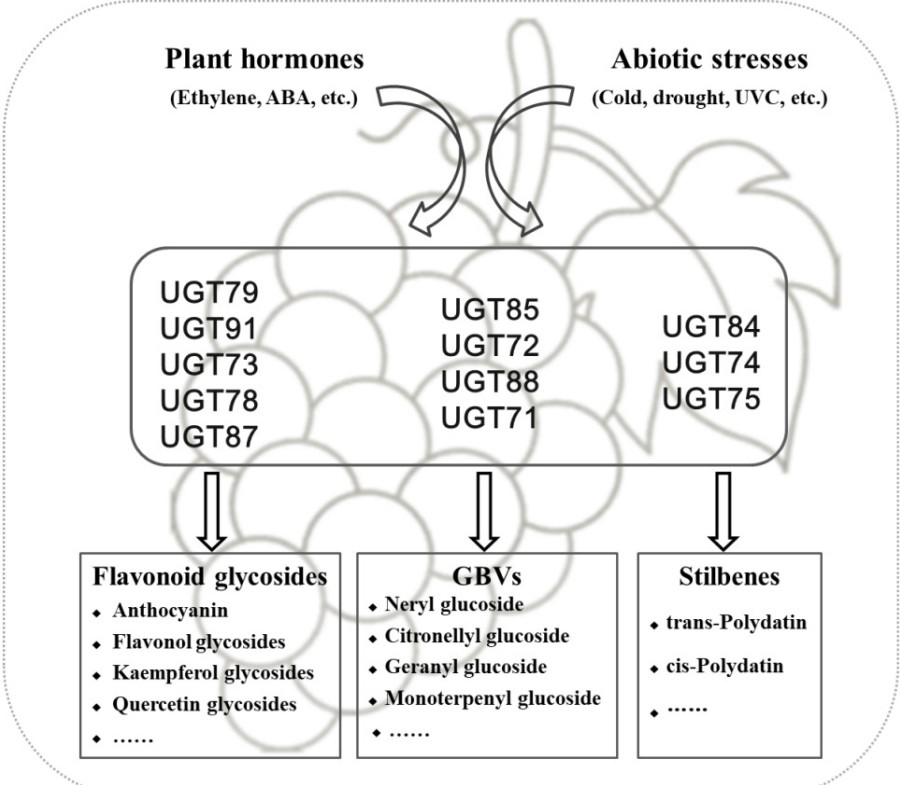

**Figure 10.** Model diagram for the synthesis and regulation of secondary metabolites related to fruit quality involved in UGT in grape fruits. When fruits are developed to a certain stage, the expression of different UGT members can be activated under the co-regulation of endogenous plant hormones and external environmental signals, thereby affecting the accumulation of secondary metabolites such as flavonoid glycosides, GBVs and stilbenes.

## 5. Conclusions

The UGT gene family in grapes contains 228 members divided into 15 groups and is distributed unevenly on 18 chromosomes. All VvUGT members have UDPGT conserved domains, 57.9% of which have lost introns, and 186 VvUGT member sequences contain 5′-UTR or 3′-UTR. The promoter sequences of *VvUGT* contain cis-acting elements related to plant hormone, plant growth and development, and stress response, and 12 groups of UGT members respond to UVC, low temperature, drought, and other external stresses to varying degrees. More than 72% of *UGT* members are expressed at one or more stages of grape fruit development, and the expression levels of 34 *UGT* members increase gradually with the increase of fruit maturity. The UGT members of different groups may be involved in the accumulation of secondary metabolites such as flavonoids, GBVs, and stilbenes. These results will provide a new entry point for further research on the function and regulation mechanism of UGT genes, which is closely related to the formation of grape fruit quality, thereby providing a new idea for the improvement of fruit quality and cultivation facilities.

**Supplementary Materials:** The following are available online at https://www.mdpi.com/article/1 0.3390/horticulturae7080204/s1, Table S1: Primers of candidate *VvUGT* genes and reference gene used for qRT-PCR, Table S2: Information of UGT genes in grape.

**Author Contributions:** Conceptualization, L.W. and S.L.; methodology, Y.W. (Yongzan Wei); software, Y.W. (Yi Wang) and H.M.; validation, Y.W. (Yongzan Wei) and G.X.; formal analysis, Y.W. (Yongzan Wei) and Y.W. (Yi Wang); investigation, H.M. and Y.L.; resources, L.W.; data curation, Y.W. (Yongzan Wei) and H.M.; writing—original draft preparation, Y.W. (Yongzan Wei) and L.W.; writing—review and editing, Y.W. (Yongzan Wei), H.M. and L.W.; visualization, Y.W. (Yi Wang); supervision, S.L.; project administration, L.W.; funding acquisition, L.W. All authors have read and agreed to the published version of the manuscript.

**Funding:** This research was funded by the National Key R&D Program of China (Grant No. 2019YFD1000101), Strategic Priority Research Program of the Chinese Academy of Sciences (Grant No. XDA23080602), and the National Natural Science Foundation of China (Grant No. 31672120).

**Institutional Review Board Statement:** Not applicable.

**Informed Consent Statement:** Not applicable.

**Data Availability Statement:** The data supporting the findings of this study are available from the corresponding author upon reasonable request.

**Conflicts of Interest:** The authors declare no conflict of interest.

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
