# Peer review of "Genome-Wide Analysis and Functional Characterization of the UDP-Glycosyltransferase Family in Grapes"

_horticulturae, doi:10.3390/horticulturae7080204_

Round 1

Reviewer 1 Report

Dear Authors,

the manuscript: "Genome-wide Analysis and Functional Characterization of the UDP-glycosyltransferase Family in Grapes" made a very good impression on me. The presented results of the study of the functions and mechanisms of gene regulation in grapevines are currently one of the most modern and very popular trends in research. The following are some notes to improve this manuscript:

1) Keywords should not repeat words from the title of the manuscript.

2) In Introduction, the hypothesis and purpose of the study should be better expressed.

3) The research methodology does not raise my objections. L.70 - use the full Latin name.

4) The results are well presented, but my reservations are raised by the graphic form of their presentation - most of the Figures are illegible: Figure 1 - please enlarge, Figures 3-7 - are illegible to varying degrees, you need to enlarge these Figures to the size of the page.

5) Table 1 - adjust to TEMPLATE.

6) In addition, the entire manuscript must be re-analyzed in terms of compliance with the template adopted for Horticulturae.

I believe that the research presented in this paper has high scientific value and the manuscript is carefully prepared. After considering the above-mentioned comments, the manuscript deserves to be published in the journal Horticulturae.

Author Response

Response to Reviewer 1 Comments

Point 1: 1) Keywords should not repeat words from the title of the manuscript.

Response 1: We have selected different keywords according to the reviewer’s suggestion.

Point 2: 2) In Introduction, the hypothesis and purpose of the study should be better expressed.

Response 2: We have re-written this part to further improve it. Thank you for the suggestion.

Point 3: 3) The research methodology does not raise my objections. L.70 - use the full Latin name.

Response 3: We have revised according to the reviewer’s suggestion.

Point 4: 4) The results are well presented, but my reservations are raised by the graphic form of their presentation - most of the Figures are illegible: Figure 1 - please enlarge, Figures 3-7 - are illegible to varying degrees, you need to enlarge these Figures to the size of the page.

Response 4: We have resized figures and added image definition according to the reviewer’s suggestion.

Point 5: 5) Table 1 - adjust to TEMPLATE.

Response 5: We have revised Table 1 refer to TEMPLATE.

Point 6: 6) In addition, the entire manuscript must be re-analyzed in terms of compliance with the template adopted for Horticulturae.

Response 6: Refer to the template adopted for Horticulturae, we have rechecked carefully the full text and revised all format and clerical errors in the manuscript.

Reviewer 2 Report

The paper "Genome-wide Analysis and Functional Characterization of the UDP-glycosyltransferase Family in Grapes" is well organized and it a significant contribution to the field. I found it original, well written, scientific sounding. 

Minor comments:

Line 141: please, correct  “from pineapple tissues by using a RN40

Line 153 and throughout the paper: Arabidopsis in italic

Line 181-183: paragraph not understandable

Fig. 5: improve the legend explaining better the figure

Please, describe better the legend for Table S2 Information of UGT family from grape

Lines 416-419: Please, remove

Lines 543-544: Please, remove

Author Response

Response to Reviewer 2 Comments

Point 1: Line 141: please, correct “from pineapple tissues by using a RN40

Response 1: We have modified this error according to the reviewer’s suggestion.

Point 2: Line 153 and throughout the paper: Arabidopsis in italic

Response 2: We have rechecked carefully the full text and revised these errors according to the reviewer’s suggestion.

Point 3: Line 181-183: paragraph not understandable

Response 3: We have revised this paragraph according to the reviewer’s suggestion.

Point 4: Fig. 5: improve the legend explaining better the figure

Response 4: We have re-written the legend description in Fig. 5 according to the reviewer’s suggestion.

Point 5: Please, describe better the legend for Table S2 Information of UGT family from grape

Response 5: We have re-written the legend description in Table S2 according to the reviewer’s suggestion.

Point 6: Lines 416-419: Please, remove

Response 6: We have removed this part according to the reviewer’s suggestion.

Point 7: Lines 543-544: Please, remove.

Response 7: We have removed this part according to the reviewer’s suggestion.
